# Prevalence of Selected Polymorphisms of Il7R, CD226, CAPSL, and CLEC16A Genes in Children and Adolescents with Autoimmune Thyroid Diseases

**DOI:** 10.3390/ijms25074028

**Published:** 2024-04-04

**Authors:** Hanna Borysewicz-Sańczyk, Natalia Wawrusiewicz-Kurylonek, Joanna Gościk, Beata Sawicka, Filip Bossowski, Domenico Corica, Tommaso Aversa, Małgorzata Waśniewska, Artur Bossowski

**Affiliations:** 1Department of Pediatrics, Endocrinology, Diabetology with Cardiology Divisions, Medical University of Bialystok, J. Waszyngtona 17, 15-274 Bialystok, Poland; beata.sawicka@umb.edu.pl (B.S.); bossowski.filip@gmail.com (F.B.); 2Department of Clinical Genetics, Medical University of Bialystok, J. Waszyngtona 13, 15-089 Bialystok, Poland; natalia.wawrusiewicz-kurylonek@umb.edu.pl; 3Department of Endocrinology, Diabetology and Internal Medicine, Medical University of Bialystok, M. Skłodowskiej-Curie 24A, 15-276 Bialystok, Poland; 4Faculty of Computer Science, Bialystok University of Technology, Wiejska 45A, 15-351 Bialystok, Poland; j.goscik@pb.edu.pl; 5Unit of Pediatrics, Department of Human Pathology of Adulthood and Childhood, University of Messina, Via Consolare Valeria Cap, 98125 Messina, Italy; domenico.corica@unime.it (D.C.); taversa@unime.it (T.A.); mwasniewska@unime.it (M.W.)

**Keywords:** Graves’ disease (GD), Hashimoto’s thyroiditis (HT), autoimmune thyroid diseases (AITDs), children, genetic susceptibility, single-nucleotide polymorphism (SNP), Il7R, CD226, CAPSL, CLEC16A

## Abstract

Hashimoto’s thyroiditis (HT) and Graves’ disease (GD) are common autoimmune endocrine disorders in children. Studies indicate that apart from environmental factors, genetic background significantly contributes to the development of these diseases. This study aimed to assess the prevalence of selected single-nucleotide polymorphisms (SNPs) of Il7R, CD226, CAPSL, and CLEC16A genes in children with autoimmune thyroid diseases. We analyzed SNPs at the locus rs3194051, rs6897932 of IL7R, rs763361 of CD226, rs1010601 of CAPSL, and rs725613 of CLEC16A gene in 56 HT patients, 124 GD patients, and 156 healthy children. We observed significant differences in alleles IL7R (rs6897932) between HT males and the control group (C > T, *p* = 0.028) and between all GD patients and healthy children (C > T, *p* = 0.035) as well as GD females and controls (C > T, *p* = 0.018). Moreover, the C/T genotype was less frequent in GD patients at rs6897932 locus and in HT males at rs1010601 locus. The presence of the T allele in the IL7R (rs6897932) locus appears to have a protective effect against HT in males and GD in all children. Similarly, the presence of the T allele in the CAPSL locus (rs1010601) seems to reduce the risk of HT development in all patients.

## 1. Introduction

Loss of immune tolerance to tissue-specific antigenic peptides underlies autoimmune diseases and leads to an immune response directed against the body’s own cells. Complex immune mechanisms, including immune system dysfunction, are involved in the pathogenesis of autoimmune diseases [1]. Common chronic autoimmune endocrine disorders in children include autoimmune thyroid diseases (AITDs), such as Hashimoto’s thyroiditis (HT) and Graves’ disease (GD) [2,3]. The mechanisms leading to the development of these diseases remain unknown; however, scientific reports indicate that besides environmental factors, genetic background plays a significant role [3,4,5]. The genetic predisposition to AITDs has been attributed to approximately 70% of disease risk, while environmental factors are believed to trigger the disease in genetically susceptible individuals [3]. A number of genetic determinants of AITDs have already been established through candidate gene studies; however, the contribution of particular genes is not significant, and it is possible that polymorphisms play a role in multiple genes [3]. 

Single-nucleotide polymorphisms (SNPs) are the simplest and most common form of DNA variation occurring throughout the genome. They can be responsible for variations among individuals, the evolution of the genome, familial traits, or inter-individual differences in responses to drugs and susceptibility to diseases. Identifying gene variants and analyzing their effects can lead to a better understanding of their impact on gene function and individual health [6,7]. This knowledge can provide a starting point for the development of new, useful markers for disease detection and therapeutic approaches.

In our previous studies, we demonstrated that some polymorphisms of the genes for IL2RA, FAIM2, IFIH1, PADI4, or CTLA-4 appeared more frequently in children and adolescents with autoimmune diseases such as type 1 diabetes (T1D) and AITDs, which may be related to the occurrence and course of the disease [8,9]. According to the literature, there are numerous other factors involved in immune mechanisms which may potentially affect the development of autoimmune diseases. These include Il-7 and its receptor Il7R, CD226, CAPSL, and CLEC16A, which appear to be attractive candidates for further studies to explore the pathogenic mechanisms and potential therapeutic options for autoimmune diseases.

The cytokine IL7 and its receptor, IL7R, are essential for T and B cell development, as well as the differentiation and survival of naive T cells and the generation and maintenance of memory T cells [10,11]. Stimulation of the receptor for Interleukin 7 (IL7R) has been shown to play an important role in the development and progression of autoimmune diseases [12,13,14,15]. The IL7R gene has been identified as one of the key genes and pathways associated with the development of Hashimoto’s thyroiditis [16]. Furthermore, recent studies suggest that effects on the IL7/IL7R pathway may be important in the treatment of autoimmune diseases [17,18]. The IL7R is a heterodimer consisting of a common gamma chain (encoded by IL2RG) and a specific alpha chain (encoded by IL7R). Whereas the gamma chain is produced by most hemopoietic cells and is shared by the receptors of several cytokines (IL-2, IL-4, IL-7, IL-9, IL-15, and IL-21), the alpha chain is almost exclusively produced by lymphoid lineage cells and is required for the development and maintenance of the immune system [19].

Adjacent to the IL7R gene, another gene is located on chromosome 5p13–CAPSL (Calcyphosine-like, also known as Q8WWF8). Studies indicate a link between CAPSL and the development of autoimmune diseases such as type 1 diabetes [19]. The function of CAPSL is still unknown; however, it was established that it contains two calcium-binding motifs (EF-hands) also found in a superfamily of calcium sensors and calcium signal modulators [19].

CD226 (also known as DNAM-1) is a membrane protein involved in cell adhesion, activation, and differentiation that has been shown to be expressed on the surface of immune cells including natural killer (NK) cells, CD4+ and CD8+ T cells, monocytes, platelets, and B18 cells. Polymorphisms in the gene encoding its molecule (located at chromosome 18q22.3) have been linked to a risk factor for autoimmune diseases like T1D and Graves’ disease [20,21,22,23,24]. The possibility of CD226 affecting the treatment of autoimmune diseases has also been suggested [25]. 

CLEC16A (C-Type Lectin Domain family 16A, previously known as KIAA0350) at chromosome 16p13 has been proposed as a gene region that may play a role as a susceptibility locus for autoimmune disease. Some SNPs within the CLEC16A gene have been shown to be associated with several autoimmune diseases, such as T1D, primary adrenal insufficiency, juvenile idiopathic arthritis, celiac disease, and AITDs, among others [26,27,28]. Thus far, little is known about the detailed function of the CLEC16A gene product. It is highly expressed in B lymphocytes, NK cells, and dendritic cells (DC), and is thought to be involved in providing a signal for immune tolerance [29]. 

The aim of this study was to assess the prevalence of selected single-nucleotide polymorphisms (SNPs) of Il7R, CD226, CAPSL, and CLEC16A genes in children and adolescents with autoimmune thyroid diseases, and in the control group. We compared these data in an attempt to demonstrate which of the studied SNPs may play a role in susceptibility to disease or show a protective effect against autoimmune thyroid disease in children and adolescents.

## 2. Results

We included in the study 56 patients with HT (mean age, 15.2 ± 2.2 years), 124 patients with GD (mean age, 16.5 ± 2 years), and 156 healthy, euthyroid volunteers (mean age, 16.3 ± 3 years) of comparable age, weight, and height as controls. The characteristics of the study group are presented in Table 1.

The differences between AITD patients and a control group in the analyzed SNPs (rs3194051 and rs6897932 for the IL7R gene, rs763361 for the CD226 gene, and rs725613 for the CLEC16A gene) are presented in Table 2 and Table 3.

In our study, we observed a higher frequency of the C allele at the rs6897932 SNP of the IL7R gene in GD children in comparison to controls [*p* = 0.03, odds ratio (OR) 1.52 and 95% confidence interval 1.04–2.24] (Table 2). Furthermore, the C allele occurred more frequently in GD girls than in controls [*p* = 0.017, odds ratio (OR) 1.82 and 95% confidence interval 1.11–3.00]) (Table 4) and in HT boys than in the control group [*p* = 0.017, odds ratio (OR) 6.58 and 95% confidence interval 1.30–160.93] (Table 5). Moreover, the C/T genotype at the rs6897932 SNP of the IL7R gene was statistically significantly less frequent in all GD patients [*p* = 0.007, with an odds ratio (OR) of 0.51 and 95% confidence interval of 0.30–0.83] (Table 2). The data are presented in Figure 1.

The study revealed the C/T genotype at the CAPSL locus (rs1010601) in HT boys to be statistically significantly less frequent in comparison to the control group [*p* = 0.013, with an odds ratio (OR) of 0.09 and 95% confidence interval of 0.003–0.623] (Table 6 and Figure 2).

## 3. Discussion

Among the susceptibility regions for autoimmune diseases, we evaluated the polymorphisms of four genes (IL7R, CAPSL, CD226, and CLEC16A) and found the SNPs within two of these genes, IL7R and CAPSL, to be significantly associated with AITDs.

### 3.1. IL7R

The findings of our study indicate the association between the rs6897932 variant of the IL7R gene and AITDs. The C allele occurs more frequently in GD patients as well as GD girls and HT boys in comparison to the control group and seems to be the risk allele. Therefore, patients carrying the C allele are more likely to develop the disease than those carrying the T allele. Thus, the T allele appears to be a protective one. In the literature, rs6897932 polymorphism of the IL7R gene was found to be associated with susceptibility to multiple autoimmune diseases [30]. The protective effect of the T allele in rs6897932 SNP was first shown in multiple sclerosis (MS) and other autoimmune diseases including T1D [15,31,32]. Observations on the association of the protective effect of the T allele in other autoimmune diseases remain consistent with our results in patients with autoimmune thyroid diseases. However, in contrast to our study, in their work examining the T1D-associated SNPs (including rs6897932 polymorphism of IL7R gene) in individuals with Graves’ disease, Todd et al. did not find an association between this polymorphism and thyroid disease [21]. On the other hand, our study demonstrated no significant association between the other analyzed SNP of the IL7R gene (rs3194051) and AITDs in children and adolescents, even though in the study by Santiago et al., an association between this SNP and another autoimmune disease, T1D, was observed [19]. However, to the best of our knowledge, there are no other studies analyzing rs3194051 in children with AITDs.

### 3.2. CAPSL

Our results suggest that rs1010601 polymorphisms of the CAPSL gene confer susceptibility to AITDs as the C/T genotype at that locus in HT boys was statistically significantly less frequent in comparison to the control group. However, we did not observe any significance for the TT genotype (as the T allele appears to be a protective allele) probably due to an insufficiently large study group. To our knowledge, this is the first study indicating the association of rs1010601 polymorphism of the CAPSL gene with AITDs since previous works focused on the CAPSL gene SNPs in another autoimmune disease, T1D [19]. However, another CAPSL gene polymorphism (rs1445898) in the aforementioned study by Todd et al. revealed the potential association with the disease in a group of patients with GD [21]. As autoimmune thyroid diseases are known to share genetic susceptibility with T1D [21], it would be extremely interesting to confirm our observations of rs1010601 polymorphism in a larger group of patients with HT and GD.

Interestingly, CASPL and DGKA (diacylglycerol kinase alpha) share a common protein domain. According to the InterPro 2022 database and GeneMANIA, DGKA and IL7R are co-expressed with no molecular interaction [33,34]. This might indicate (for the first time in the literature) a possible shared genetic basis for both autoimmune diseases: TD1 and HT. The effects of these two molecules could be studied more in depth for functional studies in the future.

### 3.3. CD226

Despite the observed relationship between rs763361 polymorphism in the CD226 gene and multiple autoimmune diseases in other studies [20,23,24,25,35], our research did not confirm the association between this SNP and susceptibility to HT and GD in children and adolescents as there was no significant difference in the incidence of the C and T alleles between the studied group and the controls. On the contrary, in their work, Hafler et al. obtained potential evidence for the association between the rs763361 SNP and AITDs [36]. A similar association was also demonstrated in the study by Todd et al. in a group of patients with GD [21]. Interestingly, Gan et al. investigated the susceptibility of rs763361 polymorphism with autoimmune Addison’s disease. In their study, they found an association between this SNP and autoimmune polyendocrinopathy syndrome type-2 (APS2), which, in addition to Addison’s disease, AITDs and T1D are also a part of; however, they did not observe a significant association with Addison’s disease alone [37].

### 3.4. CLEC16A

In our study, we did not find significant differences in the incidence of the rs725613 variant of the CLEC16A gene in patients with autoimmune thyroid diseases and healthy children and adolescents. However, Muhali et al. observed a higher frequency of the G allele in another SNP (rs6498169) of this gene in patients with AITDs [29]. On the other hand, Awata et al. demonstrated the association between the rs2903692 SNP of this gene and T1D as well as T1D complicated with AITDs, with the G allele increasing the risk of the disease, although it was not significantly associated with AITDs alone [38]. Similar observations of the association between CLEC16A rs2903692 polymorphism and T1D as well as the positive association with the co-occurrence of thyroid autoimmunity in these patients were presented in the study by Yamashita et al. [38]. 

The present study suggests that the rs6897932 locus of the IL7R gene and rs1010601 locus of the CAPSL gene confer susceptibility to AITDs in children and adolescents. The presence of the T allele in the IL7R (rs6897932) locus appears to have a potential protective effect against GD in all children, as well as HT in boys. Similarly, the presence of allele T in the CAPSL locus (rs1010601) seems to reduce the risk of HT development in all pediatric patients, although further studies are still necessary before any definitive conclusions can be reached. The limitation of this study is the sample size, which might influence the statistical analyses. Our observations must be confirmed in studies on a larger group of pediatric patients. Confirmation of our findings in the future could contribute to the development of new methods used in the diagnosis of thyroid disease and become a potential therapeutic target in pediatric patients with AITDs. Knowing the patient’s genetic predisposition to HT or GD development could accelerate proper diagnosis and appropriate treatment to prevent complications of untreated disease. That would be extremely important in children with other coexisting autoimmune diseases like T1D.

## 4. Materials and Methods

### 4.1. Patients

Our study was conducted on a group of 56 HT patients and 124 GD patients recruited from the Endocrinology Outpatient Clinic in Bialystok and the Department of Human Pathology in Adulthood and Childhood, University of Messina. Diagnosis of AITDs was based on the Polish Endocrinology Association guidelines, which correspond with the guidelines of the European Society for Pediatric Endocrinology. Inclusion in the study was determined by medical history, physical examination, laboratory tests, and ultrasound investigations. Children with HT showed clinical and biochemical symptoms of hypothyroidism and were positive for anti-TPO and/or anti-TG autoantibodies. Patients with GD presented large goiter, hyperthyroidism in laboratory tests, and positive thyrotropin receptor antibodies (TR-Ab). All patients had no other diagnosed autoimmune diseases. HT patients were treated with L-thyroxine at a dose of 1 mcg/kg/day orally and GD children received methimazole at a dose of 0.3–1.0 mg/kg/day together with propranolol (0.5–1.0 mg/kg/day) orally. The control group consisted of 156 healthy, euthyroid volunteers with no history of HT or GD and no thyroid autoantibodies. Informed consent was given by all parents of the patients and controls as well as all children over 16 years old prior to study inclusion. The study protocol was accepted by the Local Ethical Committee at the Medical University of Bialystok and adheres to the Declaration of Helsinki.

### 4.2. Blood Analysis

Blood samples for analysis were collected from the basilic vein, fasting, in the morning. The serum concentrations of thyrotropin (TSH), free thyroxine (fT4), and free triiodothyronine (fT3) were evaluated on electrochemiluminescence “ECLIA” with the Cobas E411 analyzer (Roche Diagnostics). Normal values for TSH were 0.28–4.3 (μIU/L), 1.1–1.7 ng/dL for fT4, and 2.3–5.0 pg/mL for fT3. Antibodies against TSH-Receptor (TR-Ab), Thyroid Peroxidase (TPO), and Thyroglobulin (TG) were determined using ECLIA with Modular Analytics E170 analyzer (Roche Diagnostics). Positive titer for TR-Ab were >1.75 U/L, >34 IU/mL for anti-TPO-Ab, and >115 IU/mL for anti-TG-Ab. 

### 4.3. DNA Extraction

DNA was extracted with a classical salting-out method from the blood leukocytes. All study subjects were genotyped for SNPs at the loci rs3194051 and rs6897932 for the IL7R gene, rs763361 for the CD226 gene, rs1010601 for the CAPSL gene, and rs725613 for the CLEC16A gene. TaqMan SNP genotyping assay (Applied Biosystems, Foster City, CA, USA) was used for all genotyping. For this, polymorphisms fluorogenic TaqMan probes were used. Reactions were performed in a 7900HT fast real-time PCR system (Applied Biosystems) according to the following conditions: 10 min at 95 C for starting AmpliTaq Gold activity, 40 cycles at 95 C for 15 s, and 60 C for 1 min. As a negative control served a sample without a template. It was helpful to detect any false positive signal caused by contamination. All SNPs were analyzed in duplicates.

### 4.4. Statistical Analysis

The median unbiased estimator (mid-p) of the odds ratio, the exact confidence interval, and the associated *p*-value obtained with the mid-p method were used to determine any association between genotype or allele occurrence and a patient’s status [39]. Either parametric or non-parametric methods, according to the normality and homogeneity of variance assumptions, were used to assess whether there are statistically significant differences between groups defined by genotypes and quantitative features. The false discovery rate *p*-value adjustment method was applied due to the issue of multiple testing during the post hoc analysis [40]. As proposed in [41], measure D’ of linkage disequilibrium was used. For all calculations, a *p*-value of <0.05 was considered to be significant. The R software version 4.3.1 (16 June 2023) (Vienna, Austria) environment was utilized for all calculations [42]. Statistical power calculation with respect to the total sample size was calculated with the use of G^*^Power ver. 3.1.9.6 software [43]. Cohen’s w was applied as a measure of effect size. The Hardy–Weinberg Equilibrium was checked with the utilities of the genetics package [44].

## 5. Conclusions

To conclude, in our study we found that the presence of T allele in the IL7R (rs6897932) locus appears to have a protective effect against HT in males, as well as GD in all children. Similarly, the presence of the T allele in the CAPSL locus (rs1010601) seems to reduce the risk of HT development in all pediatric patients. On the other hand, our results seem to discard the influence in AITDs susceptibility of rs3194051 for IL7R, rs763361 for CD226, and rs725613 for CLEC16A SNPs previously associated with other autoimmune diseases. 

## Figures and Tables

**Figure 1 ijms-25-04028-f001:**
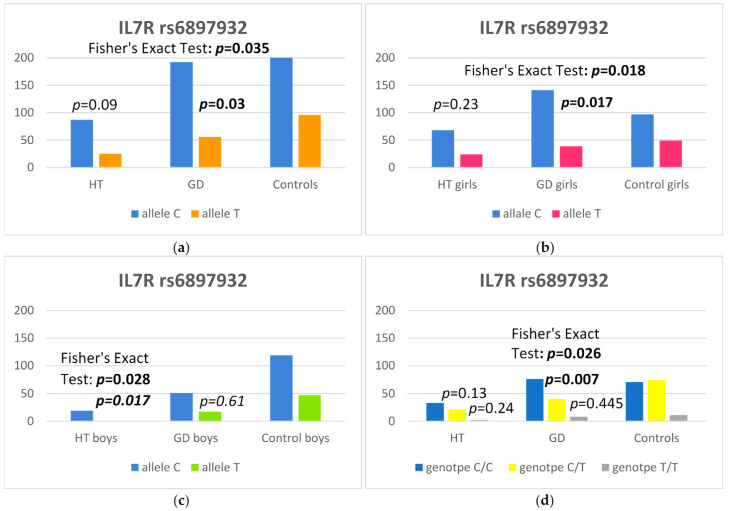
Number of alleles at rs6897932 polymorphism in the IL7R region: (**a**) in the whole study group, (**b**) in girls, (**c**) in boys and (**d**) number of genotypes at rs6897932 polymorphism in the IL7R region.

**Figure 2 ijms-25-04028-f002:**
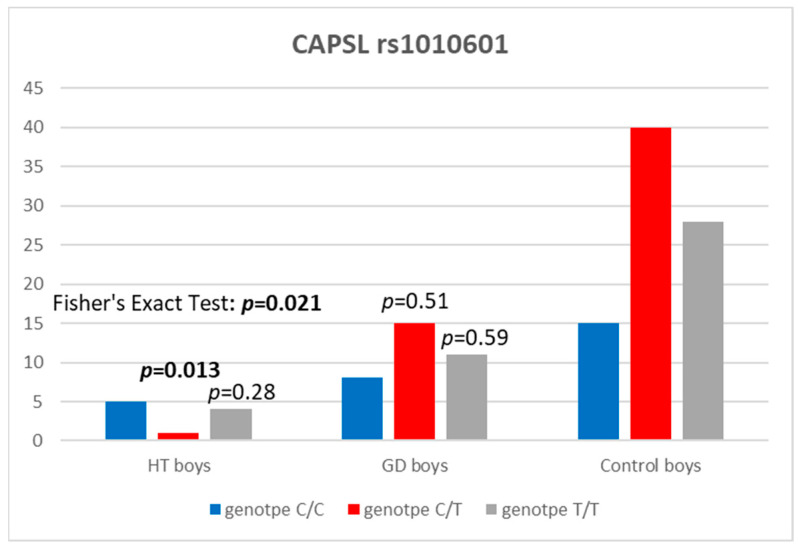
Number of genotypes at rs1010601 polymorphism in the CAPSL region in boys.

**Table 1 ijms-25-04028-t001:** Clinical characteristics of patients with HT, GD, and controls.

	HT (Mean ± SD)	*p* *	GD (Mean ± SD)	*p* **	Controls (Mean ± SD)
n (F/M)	56 (46/10)		124 (90/34)		156 (73/83)
Age (years)	15.2 ± 2.2	ns	16.5 ± 2	ns	16.3 ± 3
Weight (kg)	58 ± 5.28	ns	55.19 ± 2.39	ns	60.9 ± 7.8
Height (cm)	154.26 ± 4.14	ns	162.19 ± 2.69	ns	160 ± 8
BMI (kg/m^2^)	24.45 ± 1.33	ns	21.1 ± 2.1	0.012	23.78 ± 2.5
TSH (mIU/L)	9.87 ± 4.37	<0.025	0.37 ± 0.1	<0.01	3.04 ± 0.72
fT4 (ng/dL)	1.21 ± 0.03	ns	3.6 ± 1.4	<0.001	1.1 ± 0.17
fT3 (pg/mL)	3.08 ± 0.5	ns	7.19 ± 1.65	<0.001	3.79 ± 0.18
TR-Ab (IU/L) ^a^	0.5 ± 0.32	ns	11.56 ± 2.11	<0.001	0.4 ± 0.2
Anti-TPO Ab (IU/mL)	329.91 ± 92.93	<0.001	331.97 ± 58.12	<0.001	26.72 ± 6.8
Anti-TG Ab (IU/mL)	620.98 ± 240.34	<0.001	347.49 ± 86.7	<0.001	41.64 ± 12.1
Treatment	L-thyroxine		Methimazole		-

^a^ TR-Ab were analyzed in selected group of patients with HT (*n* = 43). ns—not statistically significant. * Statistical significance between patients with HT and controls. ** Statistical significance between patients with GD and controls.

**Table 2 ijms-25-04028-t002:** Allelic and genotypic frequencies in IL7R, CD226, CAPSL, and CLEC16A polymorphisms in GD patients.

	GD, *n* (%)	Controls, *n* (%)	OR (95% CI)	*p*
IL7R rs3194051				
A	161 (65)	203 (65)	1 (reference)	
G	87 (35)	109 (35)	1.01 (0.70–1.43)	0.971
AA	57 (46.0)	73(46.8)	1 (reference)	
AG	47 (37.9)	57 (36.5)	1.05 (0.63–1.78)	0.838
GG	20 (16.1)	26 (16.6)	0.98 (0.49–1.95)	0.969
IL7R rs6897932				
C	192 (77.4)	216 (69.2)	1.52 (1.04–2.24)	0.030
T	56 (22.6)	96 (30.8)	1 (reference)	
CC	76 (61.3)	71 (45.5)	1 (reference)	
CT	40 (32.2)	74 (47.4)	0.51 (0.30–0.84)	0.007
TT	8 (6.5)	11 (7.1)	0.68 (0.25–1.80)	0.445
CD226 rs763361				
C	136 (55.0)	173 (55.0)	1 (reference)	
T	112 (45.0)	139 (45.0)	1.02 (0.73–1.43)	0.88
CC	43 (34.7)	53 (34.0)	1 (reference)	
CT	50 (40.3)	67 (43.0)	0.92 (0.53–1.59)	0.765
TT	31 (25.0)	36 (23.0)	1.06 (0.56–1.99)	0.853
CAPSL rs1010601				
C	111 (45.0)	123 (39.4)	1.24 (0.89–1.75)	
T	137 (55.0)	189 (60.6)	1 (reference)	0.205
C	1 (0.8)	0 (0)		
CC	31 (25.0)	28 (17.9)	1.1 (0.03–44.50)	0.950
CT	47 (37.9)	67 (42.9)	0.7 (0.02–27.92)	0.827
TT	45 (36.3)	61 (39.1)	0.7 (0.02–29.38)	0.851
CLEC16A rs725613				
G	79 (31.9)	101 (32.4)	1 (reference)	
T	169 (68.1)	211 (67.6)	1.02 (0.72–1.47)	0.898
GG	10 (8.0)	15 (9.6)	1 (reference)	
GT	59 (47.6)	71 (45.5)	1.24 (0.52–3.07)	0.632
TT	55 (44.3)	70 (44.9)	1.17 (0.49–2.91)	0.724

**Table 3 ijms-25-04028-t003:** Allelic and genotypic frequencies in IL7R, CD226, CAPSL, and CLEC16A polymorphisms in HT patients.

	HT, *n* (%)	Controls, *n* (%)	OR (95% CI)	*p*
IL7R rs3194051				
A	72 (64.3)	203 (65.1)	1 (reference)	
G	40 (35.7)	109 (34.9)	1.03 (0.65–1.62)	0.88
AA	24 (42.8)	73 (46.8)	1 (reference)	
AG	24 (42.8)	57 (36.5)	1.28 (0.65–2.50)	0.47
GG	8 (14.3)	26 (16.7)	0.94 (0.35–2.32)	0.90
IL7R rs6897932				
C	87 (77.7)	216 (69.2)	1.5 (0.94–2.60)	0.09
T	25 (22.3)	96 (30.8)	1 (reference)	
CC	33 (59.0)	71 (45.5)	1 (reference)	
CT	21 (37.5)	74 (47.4)	0.61 (0.32–1.16)	0.13
TT	2 (3.5)	11 (7.1)	0.42 (0.06–1.70)	0.24
CD226 rs763361				
C	51 (45.5)	173 (55.4)	1 (reference)	
T	6154.5)	139 44.6)	1.49 (0.96–2.30)	0.07
CC	11 (19.6)	53 (34.0)	1 (reference)	
CT	29 (51.8)	67(42.9)	2.06 (0.96–4.70)	0.06
TT	16 (28.6)	36 (23.1)	2.12 (088–5.26)	0.09
CAPSL rs1010601				
C	45 (40.2)	123 (39.4)	1.03 (0.66–1.60)	0.89
T	67 (59.8)	189 60.6)	1 (reference)	
CC	11 (19.6)	28 (17.9)	1 (reference)	
CT	23 (41.1)	67 (42.9)	0.87 (0.38–2.09)	0.75
TT	22 (39.3)	61 (39.1)	0.91 (0.39–2.21)	0.84
CLEC16A rs725613				
G	31 (27.7)	101 (32.4)	1 (reference)	
T	81 (72.3)	211 (67.6)	1.25 (0.78–2.03)	0.36
GG	3 (5.4)	15 (9.6)	1 (reference)	
GT	25 (44.6)	71 (45.5)	1.69 (0.50–8.13)	0.42
TT	28 (5.0)	70 (44.9)	1.92 (0.57–9.19)	0.31

**Table 4 ijms-25-04028-t004:** Allelic and genotypic frequencies in rs6897932 polymorphisms of IL7R gene in GD girls.

IL7R rs6897932	GD Girls, *n* (%)	Controls, *n* (%)	OR (95% CI)	*p*
C	141 (78.3)	97 (66.4)	1.82 (1.11–3.00)	0.017
T	39 (21.7)	49 (33.6)	1 (reference)	
CC	56 (62.3)	30 (41.1)	1 (reference)	
CT	29 (32.2)	37 (50.7)	0.42 (0.21–0.81)	0.010
TT	5 (5.5)	6 (8.2)	0.45 (0.12–1.67)	0.223

**Table 5 ijms-25-04028-t005:** Allelic and genotypic frequencies in rs6897932 polymorphisms of IL7R gene in HT boys.

IL7R rs6897932	HT Boys, *n* (%)	Controls, *n* (%)	OR (95% CI)	*p*
C	19 (95)	119 (71.7)	6.58 (1.30–160.93)	0.017
T	1 (5)	47 (28.3)	1 (reference)	
CC	9 (90)	41 (49.4)	1 (reference)	
CT	1 (1)	37 (44.6)	0.14 (0.005–0.82)	0.026
TT	0 (0)	5 (6)	0.99 (0.03–7.68)	0.99

**Table 6 ijms-25-04028-t006:** Allelic and genotypic frequencies in rs1010601 polymorphisms of CAPSL gene in HT boys.

CAPSL rs1010601	HT Boys, n (%)	Controls, n (%)	OR (95% CI)	*p*
C	11 (55)	70 (42.2)	1.66 (0.65–4.40)	0.29
T	9 (45)	96 (57.8)	1 (reference)	
CC	5 (50)	15 (18.1)	1 (reference)	
CT	1 (10)	40 (48.2)	0.086 (0.003–0.623)	0.013
TT	4 (40)	28 (33.7)	0.44 (0.09–1.97)	0.279

## Data Availability

Data will be available upon request.

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
