# Peer review of "Prevalence of Selected Polymorphisms of Il7R, CD226, CAPSL, and CLEC16A Genes in Children and Adolescents with Autoimmune Thyroid Diseases"

_ijms, 2024, doi:10.3390/ijms25074028_

Round 1

Reviewer 1 Report

Comments and Suggestions for Authors

The aim of this study was to assess the prevalence of a series of selected single nucleotide polymorphisms (SNPs) of Il7R, CD226, CAPSL and CLEC16A genes in children and adolescents with autoimmune thyroid diseases. Furtheron, the authors aimed to investigate if the studied SNPs play a role in the susceptibility to disease or by contrast show a protective effect against autoimmune thyroid disease in children and adolescents. The article addresses an interesting subject in thyroid pathology, it’s a complex, well-designed study; however, it needs some revisions before being accepted for publishing.

-          The aim of the study should be clearer; the authors investigated the prevalence of these SNPs in the study population (main objective), but than the aim becomes confusing; it is not mentioned anything about the control group, that the SNPs were evaluated in patients with the disease, but also in the group of control patients and that the results were compared. This is essential for the clarity of the study.

-          The conclusion part is too long, the authors should move the limitations of the study in the discussion part. Also, it is mentioned here, but very briefly about the clinical implications of the results of the study. In my opinion this data should be detailed more, in a larger paragraph in the Discussion part (how this could be implemented in every-day practice, ect).

-          The Discussion part should contain a paragraph at the end that integrates data of all SNPs of the studied genes with relevant results, completed by the discussion on the clinical relevance of these results and limitations of the study.

-                      I don’t have expertise to evaluate the accuracy of the genetic data.

-          In Material and Methods there is a paragraph regarding the data of the patients; some of the data presented and Table 3 (Table 3. Clinical characteristics of patients with HT, GD and controls) also should be moved to the Results.

Comments on the Quality of English Language

Minor editing of English language required

Author Response

Responses to Reviewers’ Comments

Manuscript ID: ijms-2900635

Title: Prevalence of selected polymorphisms of Il7R, CD226, CAPSL and CLEC16A genes in children and adolescents with autoimmune thyroid diseases

Authors: Hanna Borysewicz-SaÅ„czyk, Natalia Wawrusiewicz-Kurylonek, Joanna GoÅ›cik, Beata Sawicka, Filip Bossowski, Domenico Corica, Tommaso Aversa, MaÅ‚gorzata WaÅ›niewska and Artur Bossowski

Section: Molecular Oncology

Special Issue: Thyroid Disease and Thyroid Cancer 2.0

Comments and Suggestions for Authors

The aim of this study was to assess the prevalence of a series of selected single nucleotide polymorphisms (SNPs) of Il7R, CD226, CAPSL and CLEC16A genes in children and adolescents with autoimmune thyroid diseases. Furtheron, the authors aimed to investigate if the studied SNPs play a role in the susceptibility to disease or by contrast show a protective effect against autoimmune thyroid disease in children and adolescents. The article addresses an interesting subject in thyroid pathology, it’s a complex, well-designed study; however, it needs some revisions before being accepted for publishing.

  1. The aim of the study should be clearer; the authors investigated the prevalence of these SNPs in the study population (main objective), but than the aim becomes confusing; it is not mentioned anything about the control group, that the SNPs were evaluated in patients with the disease, but also in the group of control patients and that the results were compared. This is essential for the clarity of the study.

Thank you for that remark. This is a valuable suggestion. The data were added.

  1. The conclusion part is too long, the authors should move the limitations of the study in the discussion part. Also, it is mentioned here, but very briefly about the clinical implications of the results of the study. In my opinion this data should be detailed more, in a larger paragraph in the Discussion part (how this could be implemented in every-day practice, ect).

Thank you for that valuable suggestion. The data were added in to discussion part.

  1. The Discussion part should contain a paragraph at the end that integrates data of all SNPs of the studied genes with relevant results, completed by the discussion on the clinical relevance of these results and limitations of the study.

Thank you for that suggestion. This is a valuable suggestion. The data were added.

  1. I don’t have expertise to evaluate the accuracy of the genetic data.

  1. In Material and Methods there is a paragraph regarding the data of the patients; some of the data presented and Table 3 (Table 3. Clinical characteristics of patients with HT, GD and controls) also should be moved to the Results.

Thank you for that suggestion. The table 3 and some data waere moved to the Results Section.

Comments on the Quality of English Language

  1. Minor editing of English language required

Thank you for that remark. The language was corrected.

Reviewer 2 Report

Comments and Suggestions for Authors

I congratulate the authors on their study. They have evaluated particular gene allies by comparing SNPs from autoimmune disease patients such as TD1, HT, and GD. However, improving the text at some points will increase the quality of the paper. 

To make the results section more understandable, prioritize your results with tables. Then, direct the flow towards only IL7R and CAPSL, where you will find significance. Doing the same order in the discussion section will enable you to create a pattern regarding the process leading to the conclusion and the CAPSL-IL7R you focus on.

Add a datum representing the n for each group to the figure legends.

The discussion paragraph should be placed after the results.

Table 3 must be in the results section.

Does this table, in which you have added many biochemical parameters, affect your results? Are the differences in IL7R you observed in HT and GD patients related to biochemical analyses? Did the authors evaluate the positive or negative correlation between parameters?

Or are data shared only for diagnostic accuracy purposes?

I recommend arranging Figures 1, 2, 3, and 4 side by side and one under the other as a single figure. With this form, it could be a reader-friendly presentation. In addition, it would be beneficial to equalize the y-axis to clarify all compared parameters for IL7R.

According to the literature, CASPL and DGKA share a common protein domain. DGKA and IL7R are co-expressed (database: InterPro in 2022. doi: 10.1093/nar/gkac993). There is no molecular interaction based on the databases. In this sense, your study will contribute new data to the literature on TD1 and HT. You can add this information provided by the InterPro database and schematized in GeneMANIA to the discussion section of your article. The effects of these two molecules can be studied in more depth for functional studies.

Make one or two sentences in the discussion section more confident. Instead of "However, the main limitation of our study, ...................." -->The limitation of this study is the sample size, which might influence the statistical analyses. Our observations must be confirmed in studies on a larger group of pediatric patients.

Comments on the Quality of English Language

Minor editing of English language required. 

Author Response

Responses to Reviewers’ Comments

Manuscript ID: ijms-2900635

Title: Prevalence of selected polymorphisms of Il7R, CD226, CAPSL and CLEC16A genes in children and adolescents with autoimmune thyroid diseases

Authors: Hanna Borysewicz-SaÅ„czyk, Natalia Wawrusiewicz-Kurylonek, Joanna GoÅ›cik, Beata Sawicka, Filip Bossowski, Domenico Corica, Tommaso Aversa, MaÅ‚gorzata WaÅ›niewska and Artur Bossowski

Section: Molecular Oncology

Special Issue: Thyroid Disease and Thyroid Cancer 2.0

Comments and Suggestions for Authors

I congratulate the authors on their study. They have evaluated particular gene allies by comparing SNPs from autoimmune disease patients such as TD1, HT, and GD. However, improving the text at some points will increase the quality of the paper.

  1. To make the results section more understandable, prioritize your results with tables. Then, direct the flow towards only IL7R and CAPSL, where you will find significance. Doing the same order in the discussion section will enable you to create a pattern regarding the process leading to the conclusion and the CAPSL-IL7R you focus on.

Thank you for that suggestion. The tables were presented at the beginning of the results and then Il7R and CAPSL significances were presented. Similarly, the discussion started with short paragraph concerning each analysed SNP to focus at the end on the (a paragraph in the discussion section added with conclusions on Il7R and CAPSL).

  1. Add a datum representing the n for each group to the figure legends.

Thank you for that suggestion. The data on the n of each group were presented in the tables.

  1. The discussion paragraph should be placed after the results.

Thank you. The discussion paragraph is placed after the results.

  1. Table 3 must be in the results section.

Thank you for that suggestion. The table 3 was moved to the results section.

  1. Does this table, in which you have added many biochemical parameters, affect your results? Are the differences in IL7R you observed in HT and GD patients related to biochemical analyses? Did the authors evaluate the positive or negative correlation between parameters? Or are data shared only for diagnostic accuracy purposes?

Thank you for that remark. The data were presented to characterise the study and control group as well as for diagnostic accuracy purposes. The correlation between parameters was not evaluated.

  1. I recommend arranging Figures 1, 2, 3, and 4 side by side and one under the other as a single figure. With this form, it could be a reader-friendly presentation. In addition, it would be beneficial to equalize the y-axis to clarify all compared parameters for IL7R.

Thank you for that remark. This is a valuable suggestion. The figures were placed side by side and one under the other as a single figure. The y-axis was equalized.

  1. According to the literature, CASPL and DGKA share a common protein domain. DGKA and IL7R are co-expressed (database: InterPro in 2022. doi: 10.1093/nar/gkac993). There is no molecular interaction based on the databases. In this sense, your study will contribute new data to the literature on TD1 and HT. You can add this information provided by the InterPro database and schematized in GeneMANIA to the discussion section of your article. The effects of these two molecules can be studied in more depth for functional studies.

Thank you for that suggestion. This is a very interesting suggestion. The information was added to the discussion section.

  1. Make one or two sentences in the discussion section more confident. Instead of "However, the main limitation of our study, ...................." -->The limitation of this study is the sample size, which might influence the statistical analyses. Our observations must be confirmed in studies on a larger group of pediatric patients.

Thank you for that remark. The sentences were improved.

Comments on the Quality of English Language

  1. Minor editing of English language required.

Thank you for that remark. The language was corrected.

Round 2

Reviewer 2 Report

Comments and Suggestions for Authors

Recommendation: accept